# Targeted Alpha Therapy (TAT) with Single-Domain Antibodies (Nanobodies)

**DOI:** 10.3390/cancers15133493

**Published:** 2023-07-04

**Authors:** Kate Hurley, Meiyun Cao, Haiming Huang, Yi Wang

**Affiliations:** 1Radiobiology and Health, Canadian Nuclear Laboratories, Chalk River, ON K0J 1J0, Canada; 2Research Center, Forlong Biotechnology Inc., Suzhou 215004, China; 3Department of Biochemistry, Microbiology, and Immunology, University of Ottawa, Ottawa, ON K1N 6N5, Canada

**Keywords:** targeted alpha therapy, single-domain antibodies, nanobodies, cancer therapy, radioimmunotherapy

## Abstract

**Simple Summary:**

Targeted alpha therapy (TαT) has revolutionized cancer treatment by delivering high-energy but short-range particles directly to tumor cells. The discovery of single-domain antibodies, or nanobodies, has opened new avenues for TαT. Owing to their small size, nanobodies exhibit excellent binding affinity and specificity, along with significant tumor uptake. Radiolabeled nanobodies offer numerous advantages over traditional TαT delivery vehicles and can be utilized not only for therapeutic purposes but also for cancer imaging. This review will delve into the properties of nanobodies in more detail and highlight recent studies involving nanobody-based TαT.

**Abstract:**

The persistent threat of cancer necessitates the development of improved and more efficient therapeutic strategies that limit damage to healthy tissues. Targeted alpha therapy (TαT), a novel form of radioimmuno-therapy (RIT), utilizes a targeting vehicle, commonly antibodies, to deliver high-energy, but short-range, alpha-emitting particles specifically to cancer cells, thereby reducing toxicity to surrounding normal tissues. Although full-length antibodies are often employed as targeting vehicles for TαT, their high molecular weight and the presence of an Fc-region lead to a long blood half-life, increased bone marrow toxicity, and accumulation in other tissues such as the kidney, liver, and spleen. The discovery of single-domain antibodies (sdAbs), or nanobodies, naturally occurring in camelids and sharks, has introduced a novel antigen-specific vehicle for molecular imaging and TαT. Given that nanobodies are the smallest naturally occurring antigen-binding fragments, they exhibit shorter relative blood half-lives, enhanced tumor uptake, and equivalent or superior binding affinity and specificity. Nanobody technology could provide a viable solution for the off-target toxicity observed with full-length antibody-based TαT. Notably, the pharmacokinetic properties of nanobodies align better with the decay characteristics of many short-lived α-emitting radionuclides. This review aims to encapsulate recent advancements in the use of nanobodies as a vehicle for TαT.

## 1. Introduction

Cancer is one of the deadliest diseases in the world; it is predicted that by 2040, the global burden will reach 28.0 million new cancer cases and 16.2 million deaths per year [1]. The development of more effective treatment strategies is essential in reducing the cancer burden. Traditional chemotherapy and radiotherapy have damaging effects on healthy cells and tissue surrounding the tumor, highlighting the need for targeted therapies that act specifically on diseased cells. As such, targeting cancer cells using monoclonal antibodies (mAbs) has emerged as a relatively new form of cancer treatment, with the first anti-cancer antibody, Rituximab (targeting CD20 for non-Hodgkin’s lymphoma, trade name Rituxan), being approved by the Food and Drug Administration (FDA) in 1997 [2]. Since then, 59 monoclonal antibodies (mAbs) have been approved by the FDA for clinical use against cancer, as of 31 December 2022 [3]. Antibodies may act by directly interfering with signaling pathways in tumor cells or evoking antibody-dependent cell-mediated cytotoxicity by attracting natural killer (NK) cells and macrophages [4]. Importantly, mAbs have also been used for the targeted delivery of cytotoxic drugs or particle-emitting radionuclides [4,5]. Despite the relative success of mAb-based therapies, they are limited by their large size and resultant low tumor penetration and slow blood clearance. On the other hand, single-domain antibodies (sdAbs), or nanobodies (Nbs), have gained recent attention for their ability to overcome the limitations of mAbs. Due to their significantly smaller size, Nbs are able to bind to targets that may not be accessible to mAbs, and have a much faster blood clearance, allowing for highly specific tumor targeting and reduced off-target effects. Interestingly, with the pandemic caused by severe acute respiratory syndrome coronavirus 2 (SARS-CoV-2), Nbs have become increasingly popular, as numerous studies have reported their potential as a treatment option for COVID-19 [6,7,8,9].

Targeted radionuclide therapy (TRNT), also called radioimmunotherapy (RIT), involves delivering a concentrated dose of radiation selectively to cancer cells and the tumor microenvironment. TRNT typically employs radiation with a relatively short path length, including alpha (α), beta (β), and Auger electrons [10,11]. One approach to TRNT is by conjugating the particle-emitting radionuclides to mAbs (or Nbs) to target tumor-associated antigens [12]. The application of radioimmunotherapies has been focused on the use of β particle radiation. For example, Tositumomab, a murine-derived immunoglobulin G2a (IgG2a) mAb conjugated with radioisotope iodine-131 (commercially called Bexxar), was approved by the FDA in 2003 for the treatment of relapsed non-Hodgkin lymphoma [13]. However, recently, the use of α particle-emitting radioisotopes, which can deliver relatively greater amounts of hyper-localized ionizing radiation, has attracted attention from the research community [14]. For example, radium-223 dichloride (Xofigo Injection, by Bayer HealthCare Pharmaceuticals Inc., Hanover, NJ, USA) was approved in 2013 for use in treating symptomatic bone metastases in prostate cancer patients and is considered to be the first approved targeted alpha therapy (TαT) [15]. Despite the lack of FDA-approved antibody-conjugated TαTs, many preclinical and clinical studies are currently underway to investigate their safety and efficacy [14].

## 2. Antibody Vehicles for TαT

### 2.1. Full-Length Antibodies

Mammalian blood contains five isotypes of antibodies, with IgG (immunoglobulin γ) types being the most abundant [16]. The typical structure of mammalian IgG antibodies (MW = 150 kDa) consists of two identical heavy polypeptide chains (H-chains, MW = 55 kDa each) and two identical light polypeptide chains (L-chains, MW = 25 kDa each) [17]. Together, two H-chains and two L-chains are combined to form a full antibody, which is Y-shaped in nature. Each antibody thus contains two identical fragments of each type: (1) fragment antigen-binding (Fab) fragments and (2) fragment crystallizable (Fc) regions, as indicated in Figure 1 [18]. Each antibody has the capacity to bind antigen epitopes, such as tumor-associated biomarkers, with high affinity, via the variable regions of the Fab fragments [18].

Although TαTs using mAb vehicles have shown some success in clinical trials, they are limited by complications such as myelosuppression and abnormal liver function [19,20]. These toxicities are thought to potentially be a result of the long serum half-life of mAbs (due to their high MW) and the presence of an Fc region that may interact with Fc-receptors in myeloid and hepatic sinusoidal cells [18]. Thus, researchers have worked to improve such issues by engineering smaller antibody fragments, without an Fc region, including a 25 kDa single-chain Fv (scFv), Fab (50 kDa), diabodies (55 kDa), and minibodies (80 kDa) [21]. These smaller fragments can be delivered more rapidly to the tumor, with better tumor penetration, while being cleared from circulation relatively quickly [18]. For example, one study successfully conjugated 213-Bismuth to anti-human epidermal growth factor receptor 2 (HER2) C6.5 scFv and diabody molecules, but therapeutic effects on tumors were limited, likely due to the half-life of 213-Bismuth being too short [22]. Another study also employed a C6.5 diabody (anti-HER2), but instead conjugated it to Astatine-211, allowing for the biological half-life of the delivery agent to match the physical half-life of the radioisotope [23]. The results indicated that this conjugated radionuclide can be an effective radioisotope for solid tumors, with mice showing significant delays in tumor growth, highlighting the potential use of smaller antibody fragments [23].

### 2.2. Nanobodies

The development of new molecules for the conjugation and optimization of the in vivo biodistribution of radionuclides remains at the forefront of many research projects. In 1993, a group of students from the Free University of Brussels discovered a special class of heavy chain-only antibodies (hcAbs, 90 kDa) in the blood of *Camelidae* (camels, dromedaries, llamas, alpacas, and vicunas) [24]. This class was unique in that it only required one functional heavy chain variable domain (VH domain) to bind the antigens, thus producing a functioning sdAb, named VHH [25]. In addition to the hcAbs present in the sera of *Camelidae,* some cartilaginous fish, including nurse sharks, wobbegong, and dogfish sharks, also produce functional hcAbs, named IgNARs [26]. These IgNARs contain a variable domain referred to as V-NARs, which are able to recognize and bind to antigens, and function independently; however, most studies primarily involve VHHs [26]. The sdAbs, VHH and V-NAR, are the smallest fragments retained from naturally produced immunoglobulins with a size of just 15 kDa and have been suitably termed ‘nanobodies’ (Nbs) [27,28]. Ablynx, now a part of Sanofi, is the current worldwide holder of the Nb trademark and has a patent portfolio of more than 500 patents related to Nb molecules, including their clinical use [29].

When compared to human-origin full-length antibodies, Nbs display greater binding specificity and affinity, lower immunogenicity, and higher tumor penetration [30]. The functional differences between an Nb VHH domain and a human VH domain can be attributed to structural differences, including several residue substitutions within framework two (V37F/Y, G44E, L45R, and W47G) and an extended CDR3 loop observed in many VHH domains [31,32]. The substitutions in framework two are thought to help enhance the solubility and stability of VHH domains, and therefore make the VHH domains less prone to dimerization and aggregation [27]. Meanwhile, extended CDR3 loops are thought to participate in intramolecular interactions with the VHH framework, indicating that CDR3 may act as a surrogate for the VL domain [33]. Notably, similar to conventional antibodies, VHH domains usually rely on CDR3 for interactions with antigens. Thus, an elongated CDR3 loop would provide significant versatility in its ability to bind to target molecules [33].

## 3. Nanobody Characteristics

### 3.1. High Stability

Nbs possess many favorable characteristics, as summarized in Table 1, allowing them to overcome issues faced by other radionuclide delivery vehicles. One such property is high stability. In comparison to conventional antibodies, Nbs have been shown to remain functional at high temperatures, up to 90 °C [34]. Studies have suggested that this stability at such high temperatures may be attributed to the Nbs’ ability to refold after denaturation, although this idea of reversible refolding has recently been questioned [35,36,37]. In addition to stability at high temperatures, Nbs also exhibit resistance to alkaline and acidic conditions and proteolytic stability [38,39]. In fact, Hussack et al. engineered a VHH by introducing a second disulfide bond in the hydrophobic core that was further stabilized at low pH and exhibited protease resistance (specifically to pepsin and chymotrypsin), with only minor disturbances in target binding affinities [40]. This high stability of Nbs allows for longer storage times at 4 °C and in simple buffers, indicating their ease of use [34,41].

### 3.2. Improved Antigen Access and Binding

Another important advantage of Nbs over full-length antibodies is their ability to easily access antigen binding sites. A large exposed loop extending from the VHH domain is thought to enable Nbs to use finger-like protrusions to penetrate the antigen cleft, compared to typical full-length antibodies that are unable to bind to clefts of enzymes or other buried epitopes [42,43,44]. Despite the reduced size of Nbs, studies have found that binding is comparable to two-domain fragments of classic immunoglobulins and, in fact, they appear to bind significantly more tightly [44]. Additionally, by comparing crystal structures, it was determined that Nbs exhibit much greater structural variability, contributing to their ability to achieve highly specific antigen binding [42]. Mitchell and colleagues also revealed that Nb paratopes (the antigen-binding part of an antibody) are drawn from a significantly larger number of sequence positions than those employed by classical antibodies, promoting diversity of the shape and physical properties of the antigen binding interface, and ultimately providing diverse binding specificities [42].

### 3.3. Low ‘Off-Target’ Effects and Immunogenicity

Fortunately, Nbs possess low toxicity and immune effects, limiting the risk of adverse reactions in patients. With their small size and high solubility in plasma, Nbs can be quickly cleared from the blood through renal excretion (as they are below the glomerular filtration threshold of 60 kDa), ultimately limiting off-target toxicity [45]. Although researchers have reported good tumor targeting with such a short blood half-life, others have worked to extend the half-life of Nbs, for example by fusing the Nb to albumin, to prevent such rapid clearance and allow for maximal tumor uptake [45,46].

A high degree of homology can be seen between VHH (camelid origin) and VH (human origin), likely contributing to the low immunogenicity observed with Nbs [47]. Ackaert et al. further investigated the immunogenicity risk profile of Nbs, using two different Nbs that are currently in phase II clinical trials as positron emission tomography (PET) tracers [47]. The authors demonstrated that both Nbs were taken up by dendritic cells, but showed a low capacity of Nbs to activate dendritic cells or induce T cell proliferation and that one Nb had a very low occurrence of anti-drug antibodies [47]. However, as Nbs are typically derived from camelid blood (foreign to humans), the potential to evoke an immune response remains, and, thus, researchers explored the humanization of Nbs [48]. Vincke et al. successfully developed a humanization strategy, in which the antigen-recognizing CDR3 loop of the Nb was grafted onto a humanized Nb scaffold [48]. Researchers also investigated another method where residues in framework two (at positions 49 and 50) could be humanized, in the event that the Nb scaffold may disturb proper CDR positioning [48]. Humanized Nbs exhibited no loss of targeting abilities and gained the potential to further reduce the risk of immunogenicity in humans [48]. Interestingly, an opposite process has also been used, as researchers worked to generate human hcAbs. However, these ‘human’ hcAbs have unfortunately suffered from poor solubility, likely due to the tendency for non-paired VH domains to bind to free light chains and aggregate [49]. Ultimately, isolation of VHH domains from camelids appears to be the most suitable method for efficient Nb production, which will be discussed in more detail in the section below.

## 4. Nanobody Production

### 4.1. Immune or Naïve Production

A common method for mAb and Nb production involves the development of an immune library. With respect to the immune library production of Nbs, messenger RNA (mRNA) is extracted from lymphocytes of camelids immunized with target antigens [50]. The mRNA is then reverse-transcribed into complementary DNA (cDNA) and subsequently amplified in two PCR reactions: (1) amplifying the ‘VHH-hinge’ portion and (2) a nested PCR, which amplifies the framework regions one to four of the VHH domain (Nb) only [50]. The VHH amplicons are cloned to generate a VHH library of 10^6^–10^8^ fragments, which can then be screened and selected for the production of specific Nbs [50].

Despite the relative success of using immunized camelids, immune libraries can be time-consuming and costly, and may only be of use for relatively stable proteins, because delicate targets readily unfold upon injection due to the adjuvants and camelids’ high body temperature [51,52]. Thus, the production of naïve VHH libraries has been suggested as an alternative method to overcome some of these limitations. Sabir et al. describe a method of generating a naïve library by isolating lymphocytes from a non-immunized camelid [53]. Following RNA purification from the cells, stepwise PCR amplification was conducted to recover the variable *vhh* gene for library construction [53]. However, to construct such large and diverse naïve Nb libraries, a large pool of blood is required (it is estimated that a total of more than 10 L of blood, from different animals, is needed) [50]. It remains unclear as to whether target-dedicated, immune-based library production or naïve library production of Nbs is superior.

### 4.2. Synthetic Production

To overcome the limitations faced by both immune and library production, and remove the need for animals, researchers have developed synthetic methods for Nb production [50,54]. Synthetic production is carried out entirely in vitro, under controlled experimental conditions [54]. Unlike Nbs from an animal, where each has its own framework sequence, Nbs from a synthetic library all possess the same framework region sequence [55]. Saerens et al. successfully identified a natural Nb that can effectively act as a plastic framework (that was later humanized), allowing for the exchange of antigen specificities from donor Nbs to its framework [56]. Although many researchers have since used this scaffold, Moutel et al. screened several hundred clones from immune and naïve llama VHH libraries to eventually find a very soluble and stable Nb for the creation of a universal humanized Nb library, known as the NaLi-H1 library [57]. Following the selection of a framework sequence, the hypervariable regions are then randomized, and the DNA is synthesized via PCR [55]. The achieved diversity of previously developed synthetic Nb libraries is typically around 10^9^, allowing for control of the library contents; however, libraries with a higher designed diversity can introduce more opportunities to find high-binding affinity Nbs [57,58,59]. In addition to following natural occurrences, a synthetic library can also be employed to recognize specific antigens, based on already known epitopes, using selection methods discussed below [60].

### 4.3. Nanobody Library Selection Methods

Many selection technologies can be used to retrieve antigen-specific Nbs after the development of Nb libraries. One of the most robust techniques is phage display, but researchers have also successfully employed yeast display and bacterial display methods [58,61]. Other systems such as ribosome display have been proposed, but this selection method remains technically demanding [55]. A recent technique known as NestLink has emerged, where Nb sequences are linked to barcoding peptides (flycodes) and mixed with antigens [59]. Nb–antigen complexes can then be purified through size exclusion chromatography and the barcode peptide is cleaved for identification through mass spectrometry [59]. This method also allows for the determination of binding efficiencies and may hold potential for future use in vivo [59]. Interestingly, to improve the phage display technique and increase its efficiency, Verheesen et al. developed a ‘real-time’ monitoring system. Here, screening for individual Nb clones that perform well can be completed in parallel with the selection procedure [51]. Andre provided an overview of in vivo phage display methodologies, highlighting them as a promising emerging approach for enhancing antibody targeting and improving the characteristics of drug delivery [62].

Figure 2 provides an overview of the nanobody production procedure.

### 4.4. Advantages of Nanobody Production

A key advantage of Nbs is their easy, fast, and relatively inexpensive production [28]. Compared to the production of mAbs or other types of antibody fragments, the production of Nbs is much simpler and circumvents many challenges typically encountered. Nbs only require one (matured) domain to recognize the antigen, while other antibodies and antibody-derived fragments need at least two domains that have undergone maturation together (as a VH and VL pair). The VH and VL domains must be amplified separately, as they are encoded by different gene segments, requiring a process that produces two separate libraries, resulting in numerous possible VH and VL combinations and an overall laborious process [49,63]. Meanwhile, Nbs consist of one domain and are thus encoded by a single gene, leading to a less labor-intensive process and products that bind with high affinity [28].

## 5. Radiolabeling Nanobodies

In TαT, Nbs are linked to α-emitters to facilitate the specific delivery of radiation to the desired target cells (cancer cells). To do so, Nbs must first be labeled with a radionuclide. By conjugating nanobodies to α-emitters, the Nbs serve as vehicles for precise delivery, ensuring that the radiation is concentrated on the intended cells while minimizing off-target effects. Nbs can be radiolabeled using a variety of methods, including direct labeling or indirect labeling via chelator and prosthetic groups, which will be outlined in the following subsections. It is important to note that the radiolabeling strategy used will impact both the effectiveness and potential side effects of TRNT, and, thus, the appropriate method must be selected.

### 5.1. Direct Labeling

One method is direct labeling, where the radioactive isotopes of iodine (^123^I, ^124^I, ^125^I, and ^131^I) are added to the active rings of the aromatic amino acid species in Nbs via electrophilic substitution [64,65]. The first step involves creating the iodine electrophile through the use of oxidizing agents such as chloramine T (Iodogen) or *N*-halosuccinimides [65]. Pruszynski et al. have shown success using the oxidant Iodogen and found that it minimizes any protein damage, as direct labeling is often associated with harsh conditions [66,67,68]. In these studies, Iodogen served to oxidize radioiodine, creating a positively charged iodine species, and thus increasing the efficiency of the binding to tyrosine’s electron-donating hydroxyl group on Nbs [66,67]. However, for internalizing targets such as HER2, direct radiolabeling may not be as suitable, due to the reduced accumulation of radioactivity in the cells as a result of the rapid excretion of the radiolabeled catabolites [69]. It is also important to note that the conjugation of therapeutic moieties to the Nb should be located at the opposite side of the antigen-binding location to prevent steric hindrance, and in the case that conjugation alters the binding capacity, indirect labeling using a chelator or prosthetic group should be considered instead [64].

### 5.2. Indirect Labeling with a Chelator Group

Other labeling strategies involve the use of an additional molecule to link the radionuclide to Nbs in the desired position. When indirectly labeling with a chelator group, metallic radioisotopes can be linked to an Nb using a chelating molecule to attach the radionuclide to the Nb in a distant position from the antigen-binding site [64,65]. Chelators are molecules that possess specific binding sites capable of forming stable complexes with radionuclides [64,65]. There are two mechanisms in which the bifunctional chelating agent (BFCA) can initially be conjugated to the Nb: pre-labeling and post-labeling [70]. With pre-labeling, the radiometal is complexed with a BFCA prior to interaction with the Nb, while post-labeling involves the BFCA being connected to the Nb first, followed by radiometal complexation [70]. The second strategy is more commonly used, as the BFCA–Nb complex can be stored in large quantities and subsequently used in smaller aliquots for radiolabeling [70].

A variety of chelator types have been used to radiolabel Nbs, and studies indicate that the choice of chelator has important effects on the behavior of the radiolabeled conjugate [70,71]. Thus, in the development of radiopharmaceuticals, the chelator with the most favorable characteristics must be selected. Hydrazinonicotinic acid (HYNIC), diethylenetriamine-pentaacetic acid (DTPA), tetraazacyclododecane-tetraacetic acid (DOTA), and 1,4,7-triazacyclononane-1,4,7-triacetic acid (NOTA) are some of the commonly used BFCAs for radiolabeling Nbs [64]. Chelators are covalently attached to Nbs (conjugation) via reactive electrophilic groups that react with the amino group of lysines on the Nb [65]. However, due to the presence of multiple amino acids in an Nb, there may be a lack of site-specificity, resulting in suboptimal pharmacokinetics and decreased affinity. Fortunately, strategies have been developed for the site-specific labeling of Nbs [72].

### 5.3. Indirect Labeling with a Prosthetic Group

Nanobodies can also be indirectly radiolabeled using a prosthetic group. Unlike chelators, prosthetic groups do not form stable complexes with the radiometal, but instead facilitate direct binding between the radiometal and antibody [64,65]. This approach involves the incorporation of a bifunctional prosthetic group that is responsible for radiolabeling and binding to the protein [73]. Nbs radiolabeled via prosthetic groups have shown increased intracellular retention and in vivo tumor uptake [67]. Certain biomolecules are internalized after binding to their respective receptors/antigens on the surface of tumor cells and are eventually catabolized in the lysosome [74]. When these molecules are radiolabeled, the catabolites bearing the radiolabel often wash out of the cells, ultimately decreasing the radioactive signal within tumor cells. In order to optimize the effectiveness of TRNT, it is important to maximize the extent and duration of radioactivity entrapment in cancer cells after internalization, while also maintaining a high degree of in vivo stability. Prosthetic agents can contain charged or polar moieties, carbohydrate residues, and/or amino acid peptides, generating membrane-impermeable catabolites, and thus increasing retention in the tumor cell after internalization [74].

One prosthetic group evaluated is known as *N*-succinimidyl-3-guanidinomethyl-5[^131^I]iodobenzbate (SGMIB), and when it was used to radiolabel an anti-HER2 Nb, 5F7, improved tumor targeting was observed [75]. Another study used [^131^I]SGMIB to radiolabel a different anti-HER2 Nb, 2Rs15d, and the results indicated a low toxicity profile and significant therapeutic efficacy [76]. Moreover, Vaidyanthan et al. synthesized a novel residualized prosthetic group, ^18^F-RL-I, that was used to label the 5F7 Nb, while preserving immunoreactivity and affinity for HER2 [77]. Although Nbs labeled with ^18^F-RL-I showed considerably higher tumor uptake, higher renal uptake was also observed [78,79].

Unfortunately, the recoil energy caused by the decay of alpha-emitters invariably destroys the chemical bonds between the alpha-emitter and vector (in this case, Nbs), which can lead to undesirable toxicities [10]. Thus, it is important to choose a labeling technique that not only provides metabolic stability to avoid cleavage of the radionuclide from the linked Nb, but also ensures that the conjugation is site-specific [65]. Ultimately, the choice of the radiolabeling method must be in accordance with the properties of the specific radionuclide being used, and must provide good yield, stability, and unaltered bioreactivity.

## 6. Use of Radiolabeled Nanobodies

### 6.1. Immunoscintigraphy

Immunoscintigraphy is a diagnostic imaging technique where a radiolabeled tracer is administered to a patient, usually intravenously, and the body is then scanned for radioactive emissions to provide information about the presence and nature of lesions [80,81]. Tracers typically consist of an antibody probe specific to a certain molecular (disease) target, coupled to a β- or γ-emitting radionuclide [81,82]. The concentrations of radioactive emissions can then be measured, using PET or single-photon emission computed tomography (SPECT) in combination with computed tomography (CT) or magnetic resonance imaging (MRI), to generate anatomical maps of localized disease markers within the body [83].

With the common understanding of alternative glucose metabolism in cancer tissues, PET and intravenous injections of [^18^F] fluoro-2-deoxyglucose (^18^F-FDG) are often used to measure the uptake of glucose [84]. However, this method is non-specific in that it only targets metabolically active (cancer) cells. In contrast, tracers that bind to membrane-expressed antigens (often mAbs or Nbs), allow for greater specificity and phenotypic characterization of cancer lesions throughout the body [85].

Imaging with radiolabeled Nbs may take place shortly after injections (same-day imaging), compared to full-length antibodies, where their long circulatory half-lives (days to weeks) often require patients to wait 2–4 days post-injection [86,87]. The presence of the full-length antibody-bound radioactive substance in the blood for a prolonged time can also contribute to relatively higher radiotoxicity [87]. Additionally, due to the Nbs’ small size, they have a distinct feature of penetrating dense tissues like tumors very easily, allowing for a relatively higher amount of tracer uptake [86]. Meanwhile, the large size of full-length antibodies leads to inefficient tumor penetration and incomplete visualization of the lesion [88].

### 6.2. Therapeutic Purposes

In addition to Nb-based diagnostic methods, Nb-based TRNT has clinically revolutionized the outcomes of cancer. Radiation therapy, including both external beam radiation and TRNT, is one of the three pillars of cancer therapy [89]. External beam radiation cannot be used to treat a disseminated lesion, and unfortunately causes lateral damage to healthy organs [90]. In comparison, TRNT can selectively deliver a radiation dose to cancer cells by employing radiopharmaceuticals that consist of a targeting ligand (e.g., mAbs and Nbs) and a radionuclide [91]. Three types of electron emission are currently in clinical or pre-clinical use: beta (β^−^), alpha (α), and Auger electrons; however, β^−^ and α particle-emitting radionuclides are the most widely used forms in Nb-based radiation therapy [92,93]. Varying physical properties and effects on tumors are associated with each type of radionuclide and will be discussed in more detail below.

Table 2 provides an overview of current clinical trials involving radiolabeled Nbs that are actively recruiting.

#### Radiation Types for Targeted Radionuclide Therapy

Auger electrons are low-energy electrons (1–10 keV) that are emitted during electron capture and/or internal conversion decay processes, with the potential for multiple electrons to be emitted per decay process [95]. However, Auger electrons have short path lengths (1–20 μm, less than one cell diameter) and, thus, a small range in biological tissue, making the radionuclide only effective when localized in the cell nucleus [95]. Many medical radionuclides have been identified as Auger electron-emitters, but most of these are not practical for Auger-based therapy due to incompatible half-lives and accompanying emissions [95].

Both β^−^- and α-emitting radionuclides have been used in cancer therapy. β^−^ particles travel long distances before dissipating all of their kinetic energy, and thus have a low linear energy transfer (LET) [96]. Although their long path length allows them to pass through tissues relatively easily and induce single-strand DNA breaks, neighboring healthy cells often experience toxic side effects [96,97]. Isotopes that have been used in oncology for radiation therapy as β^−^-emitters include ^186^Re, ^188^Re, ^166^Ho, ^89^Sr, ^32^P, ^153^Sm, and ^90^Y [97].

On the other hand, α particles have a short range (0.1 mm) and high LET, leading to double-strand DNA breaks, chromosomal damage, and G2 phase delay [97]. Alpha-emitting radionuclides are a very promising type of radiotherapeutic agent, and they possess key advantages over those that emit β^−^ particles or Auger electrons [98]. Some commonly used alpha-emitters are outlined in Table 3. With a short range in biological tissue, but a high LET, α particles are capable of destroying tumors with an increased relative biological effectiveness when compared to other radionuclide therapies, while causing less radiotoxicity to the healthy tissue surrounding the tumor [98,99]. Additionally, the cytotoxicity of α emissions is independent of the oxygen concentration, meaning that it is also effective in treating hypoxic (and typically radiation-resistant) tumors [99]. Importantly, when α-emitting radionuclides are targeted to specific tumor cells, they can be very effective in destroying metastases, which are difficult to treat with the currently available therapeutics [98]. Targeted alpha therapies (TαT) may augment the efficacy of immune-oncology or other anticancer agents, as α particle-induced cell death has been shown to stimulate immunogenic cell death, and may also generate antigen-specific T-cell responses, which can then be used to achieve a robust and effective anti-tumor response [12].

## 7. Targeted Alpha Therapy

Despite more than one hundred radionuclides having the ability to emit α particles as they undergo radioactive decay, the number of isotopes with the appropriate considerations for TαT suitable for clinical use in cancer treatment is limited [12]. The alpha-emitters ^223^Ra, ^225^Ac, ^211^At, ^227^Th, and ^213^Bi have all been used in clinical or pre-clinical trials for TαT. The first clinical trial employing TαT was conducted by Jurcic and colleagues in 1997 [12,19]. This study used a humanized anti-CD33 monoclonal antibody (HuM195) conjugated to the alpha-emitting isotope ^213^Bi to specifically target myeloid leukemia cells in patients who had acute myelogenous leukemia (AML) or chronic myelomonocytic leukemia [19]. The results demonstrated that 93% of evaluable patients had reductions in circulating blasts, with 78% showing a reduction in bone marrow blasts [19]. This became the first proof-of-concept study for systemic TαT in humans. Despite the safety, feasibility, and anti-leukemic effects of ^213^Bi-HuM195 in phase I and II clinical trials and the suggestion for its use as a clinical therapy, the widespread use of ^213^Bi is limited by its short half-life of about 45.6 min [102]. Therefore, Jurcic and colleagues conducted a later clinical trial that employed ^225^Ac instead of ^213^Bi conjugated to HuM195 in patients with relapsed or refractory AML [102,103]. Compared to ^213^Bi, ^225^Ac has a longer half-life of 10 days and can act as an in vivo generator of alpha particles at or within a cancer cell [102]. The results from this trial indicated that peripheral blasts were eliminated in 63% of the evaluable patients (at doses of 1 µCi/kg or more), and bone marrow blast reductions were observed in 67% of the patients [103]. Interestingly, Zalutsky and colleagues conducted a clinical study using a different α particle-emitting radionuclide, ^211^At [104]. With a half-life of 7.2 h, the authors ideated that it may be optimally suited for the molecularly targeted radiotherapy of strategically sensitive tumor sites, such as those found within the central nervous system (CNS) [104]. Indeed, Zalutsky et al. determined that treatment with ^211^At conjugated to an anti-tenascin monoclonal antibody (ch81C6) administered into the surgically created resection cavity of patients with recurrent CNS tumors resulted in a median overall survival time of 54.1 weeks, with no patients experiencing dose-limiting toxicity [104]. Moreover, Nilsson and colleagues conducted the first clinical experiment exploring the alpha-emitter ^223^Ra, where they investigated its therapeutic effects on breast and prostate cancer patients with skeletal metastases by intravenously injecting unconjugated ^223^Ra into patients [105]. The promising results indicated that more than half of the patients reported improved pain scores and that ^223^Ra may improve survival time, with limited side effects observed [105]. Since this initial work, numerous other pre-clinical and clinical studies have been conducted, advancing the field of TαT; however, the majority of these TαT studies use mAbs as the delivery vehicle, presenting challenges that could be overcome with Nbs.

### Nanobody-Based Targeted Alpha Therapy

The therapeutic potential of radiolabeled nanobodies has been studied preclinically in different tumor types with various β^−^ particle-emitters, such as ^177^Lu and ^131^I [76,106,107]. However, the low LET of β^−^ particles has led to a growing interest in radiolabeling nanobodies with α particle-emitters. One of the first published studies that employed Nbs for TαT involved ^213^Bi conjugated to an anti-PSMA (prostate-specific membrane antigen) Nb to investigate its efficacy as a potential therapeutic for prostate tumors [108]. The study was conducted after previous work indicated that ^213^Bi labeled with an anti-PSMA mAb showed promising results in vitro; however, in vivo application became difficult because the short half-life of ^213^Bi did not match the slow pharmacokinetics of the antibody [108,109,110]. Meanwhile, the optimal combination of ^213^Bi with an anti-PSMA Nb resulted in rapid tumor accumulation and produced double-strand breaks in PSMA-expressing tumor models [108]. Around the same time, Choi and colleagues published a study that evaluated the efficacy of the α-emitter ^211^At linked to an anti-HER2 Nb, 5F7, using two different residualized prosthetic groups to produce [^211^At]SAGMB-2Rs15d and *iso*-[^211^At]SAGMB-2Rs15d [75]. HER2 is commonly overexpressed in breast, ovarian, lung, and gastric cancers, and frequently results in more aggressive phenotypes and poorer prognoses [111]. It was found that anti-HER2 Nb 5F7 could be effectively labeled with ^211^At, resulting in high and prolonged tumor targeting, and rapid normal tissue clearance [75].

The interest in HER2 Nb TαT continued, and another anti-HER2 Nb, 2Rs15d, was identified. However, the 5F7 Nb appears to show a much higher tumor accumulation [75,112]. These differences are thought to be a result of the 5F7 Nb having the ability to activate the HER2 receptor upon binding and stimulate endocytosis, while the 2Rs15d binds to an epitope that does not activate the HER2 receptor [75,112]. Binding to an epitope that does not activate HER2 may actually provide a therapeutic advantage for the 2Rs15d Nb as it does not compete with trastuzumab (a commonly prescribed mAb used in the treatment of breast cancer) for HER2 binding, allowing radiolabeled 2Rs15d to be used in a combination therapy protocol [112]. Indeed, this idea was later confirmed in a study that coupled the anti-HER2 Nb 2Rs15d to different radionuclides, including ^225^Ac, for TRNT of HER2^+^ brain lesions and compared the radiolabeled Nb’s therapeutic efficacy to that of trastuzumab, a clinically approved anti-HER2 treatment [113]. The median survival of mice bearing small HER2^+^ intracranial tumors who received a combination of both trastuzumab and [^225^Ac]-2Rs15d was prolonged by 6.5 days compared to those treated with [^225^Ac]-2Rs15d alone, indicating that 2Rs15d does not compete with trastuzumab, allowing for co-administration of both therapeutics [113]. Notably, mice treated with [^225^Ac]-2Rs15d showed no significant signs of toxicity or mortality compared to control-treated animals during treatment follow-up [113]. Although previous studies had coupled 2Rs15d to the α-emitters ^225^Ac and ^211^At, Dekempeneer et al. worked to further explore coupling this Nb to the α-emitter ^213^Bi [114]. In comparison to other α-emitters, 98% of ^213^Bi decays to the α-emitter ^213^Po, which possesses a very short half-life, thus limiting migration from the target site and off-target toxicity [114]. The results indicated that [^213^Bi]Bi-DPTA-2Rs15d bound specifically to cells that express the HER2-receptor, causing a dose-dependent cytotoxic effect [114]. Further to this, in a subcutaneous xenograft model, [^213^Bi]Bi-DPTA-2Rs15d rapidly accumulated (15 min) in HER2-expressing tumors and resulted in a significantly longer median survival, of up to 80 days, compared to animals that received no treatment (56 days) [114]. Interestingly, this study also looked at the combinational effects of [^213^Bi]Bi-DPTA-2Rs15d and trastuzumab and indeed found that this led to extended mean survival of about 140 days, aligning with the previously discussed studies [112,113,114]. Taken together, these findings highlight the extensive preclinical work that has been completed and demonstrate the potential for TαT use in cancer treatment, especially for HER2^+^ tumors and metastases. Table 4 further outlines a number of TRNTs that use Nbs as the delivery vehicle.

Research involving Nbs for TαT continues to evolve and extend to numerous types of malignancies, such as multiple myeloma (MM; a hematological malignancy). Interestingly, researchers also noticed that treatment with a sdAb targeted for CSI (a marker for MM) radiolabeled with ^225^Ac resulted in increased programmed death-ligand 1 (PD-L1) expression in immune and nonimmune cells and a significant increase in CD8^+^ T-cells, implicating immune activation [116]. Further studies found that TαT using single-domain antibodies increased the production of cancer-fighting substances, such as interferon-γ, C-C motif chemokine ligand 5, granulocyte–macrophage colony-stimulating factor, and monocyte chemoattractant protein-1, in the blood and boosted the body’s own anti-cancer immune response in the tumor environment [119]. These results suggest immune-stimulating properties of TRNT and potential for TRNT to be used in combination with current immunotherapeutic approaches, such as immune checkpoint inhibitors, and overall, a promising new treatment option for multiple myeloma patients.

Unfortunately, many patients with MM will experience relapse, often as a result of residual and treatment-resistant myeloma cells [117]. However, recent evidence has suggested a potential patient-specific therapy involving Nb-based radionuclide therapy. In MM, uncontrolled proliferation of terminally differentiated plasma cells occurs, and with this comes severe B-cell suppression as well as excessive secretion of a patient-specific mAb/antibody fragment, known as the M-protein or paraprotein [117]. In some cases of MM, the paraprotein can become anchored to the surface of malignant plasma cells. As the expressed paraprotein’s sequence (referred to as the idiotype) is unique, it has been established as a valuable tumor-specific antigen for use in targeted therapies [117]. With this understanding, Puttemans et al. were able to generate model/patient-specific idiotype (Id) antibodies via immunization of llamas with the purified IgG fraction from serum taken from either 5T33MMId-bearing mice (murine multiple myeloma model) or patients with circulating paraprotein. After an extensive screening of candidates, a lead sdAb was eventually selected, 8379, for further evaluation of anti-Id TRNT. The researchers radiolabeled sdAb 8379 with β- (^177^Lu) and α-emitting (^225^Ac) radionuclides; however, this section will focus on results from TαT. The researchers observed that mice receiving [^225^Ac]-8379 showed significantly prolonged survival and delay in end-organ damage compared to control-treated animals [117]. Importantly, immunization with purified sera from the patients led to the generation of highly patient-specific sdAbs for two of the three patients, confirming the potential for generating sdAb for highly personalized targeted therapy of patients with MM [117].

## 8. Theranostics

The term ‘theranostics’ refers to the combination of agents for both diagnostic and therapeutic purposes. As discussed above, radiolabeled Nbs targeting cancer-specific markers can be traced using a PET or SPECT machine (diagnostic) and can emit short-range radiation (therapeutic) simultaneously, demonstrating their potential for theranostic uses as well. The radionuclides used in Nb theranostics can either be different or the same for both purposes (diagnostic and therapeutic); however, it is important that the radionuclides share similar pharmacokinetics and biodistribution profiles [86]. The inclusion of a diagnostic step can be of use for the selection of patients eligible for therapy, estimation of an effective dose, predicting adverse effects of therapy, and for treatment follow-up [71]. For example, Krasniqi and colleagues developed an anti-CD20 sdAb, 9079, and effectively radiolabeled it with ^68^Ga and ^177^Lu for PET imaging and targeted therapy, respectively [106]. The results indicated that sdAb 9079 radiolabeled with ^68^Ga showed specific tumor uptake, and treatment of mice with ^177^Lu-DTPA-sdAb 9079 significantly prolonged median survival compared to control groups, together indicating that radiolabeled sdAb 9079 shows promise as a theranostic drug for the treatment of CD20^+^ lymphomas [106]. Another study developed an anti-5T2MMid Nb (for multiple myeloma), R3B23, and was able to label it with ^99m^Tc for SPECT imaging and ^177^Lu for therapeutic purposes [123]. Importantly, this study showed that after treatment with ^177^Lu-R3B23, the researchers were able to image the animals using SPECT/microCT with ^99m^Tc and saw significantly lower uptake in the hearts of mice treated with ^177^Lu-R3B23, indicating a decrease in MM cells [123]. Although both of these studies used different radionuclides for imaging and therapeutic purposes, D’Huyvetter et al. radiolabeled the anti-HER2 Nb 2Rs15d with ^131^I for both diagnostic and therapeutic uses [76]. An initial scan using 2Rs15d labeled with low radioactive iodine allowed for patient selection and dosimetry calculations for subsequent treatment with [^131^I]SGMIB-2Rs15d in HER2^+^ murine xenograft models [76]. Ultimately, these studies highlight the ability of Nbs to be radiolabeled for theranostic purposes, allowing for better treatment monitoring and improved patient-specific therapeutics.

## 9. Considerations for Nanobody-Based Targeted Alpha Therapy

Despite Nbs demonstrating promising results for diagnostic and therapeutic purposes, it is crucial that some features are considered for pharmaceutical use. Although rapid clearance is a favorable feature for diagnostic purposes and to limit off-target effects, elimination of the Nb too quickly may pose challenges in therapeutic applications. As such, several strategies have been developed to extend the half-life of Nbs, including polyethylene glycol (PEG) modification and fusion to human serum albumin or Fc domains [124]. Additionally, another challenge faced as a result of the fast blood clearance of Nbs is the high accumulation in the kidneys, which could lead to resultant nephrotoxicity in TRNT applications. Interestingly, D’Huyvetter and colleagues have shown that renal retention may be dependent on the Nbs’ C-terminal residues and polarity [70]. Fortunately, researchers have proposed possible countermeasures, including co-infusion with Gelofusine and/or lysine or optimization of the amino-acid sequence of the Nb, to reduce unwanted kidney retention. Co-administration of positively charged amino acids (e.g., lysine) or Gelofusine, which competitively interacts with megalin/cubulin receptors, has long been known to reduce the renal retention of radiometal-labeled antibody fragments and peptides, and this has been further confirmed in studies involving Nbs [125]. Alternatively, the removal of charged amino-acid tags, for example, those used for purification or radiolabeling purposes, affects the polarity of Nbs, and consequently has an important impact on the degree of kidney retention. Indeed, removal of the c-terminal hexahistidine tag showed an important reduction in the renal retention of ^18^F-labeled Nbs [126].

Furthermore, it is important to discuss challenges that may be faced with α particle -emitters. The alpha-emitter ^213^Bi is promising when rapid accumulation at the diseased site is possible; however, longer-lived radionuclides are needed for less accessible solid tumors, where penetration of the radionuclide is needed for a long time. Fortunately, ^225^Ac has a much longer half-life and emits four alpha particles, allowing for delivery of a high therapeutic dose with relatively low administration. Although the four recoil daughters allow for greater emission of energy, these recoil energies can be several orders of magnitude higher, and therefore disrupt chemical bonds associated with targeting agents, allowing them to freely migrate in the body and cause harm to healthy tissue [127]. This introduces the last hurdle for all TαT—that the high energy released from the decay of alpha particles can break the bond to the linker, allowing for dissociation of the alpha–radionuclide from the targeted vehicles. To overcome such issues, selecting high-affinity and internalizing nanobodies has been suggested, as well as developing and using stronger linkers. Using a vector with the alpha-emitter that is internalized into the cell usually allows for the recoiling daughter radionuclides to remain inside the target cells and thus limit exposure to other tissues, with the remaining not adsorbed part of the radio conjugate being excreted from the body [128]. Another approach is to inject the α-emitting radionuclides in or near the tumor tissue (or in the cavity after tumor resection), as tested by Cordier et al. and Krolicki et al. by locally injecting gliomas [129,130]. A third method for reducing the recoil problem is to encapsulate the mother radionuclide in a nanoparticle that is able to confine all recoils in the decay chain within its structure [131]. These nanoparticles involve structures such as liposomes and polymersomes and should allow adequate surface functionalization to enable systemic administration and efficient tissue targeting [131]. The type of material, size, and shape of various nanoparticles that can be used to encapsulate α-emitters has been extensively reviewed by previous groups [131,132].

Finally, the production and preparation of α-emitting radionuclides can be a challenging aspect for TαT industrialization. Medical isotope shortages are a concern globally due to limited source material and difficult production processes. Alpha radionuclides also have relatively short half-lives, adding to logistical considerations. Currently, ^211^At and ^225^Ac are available in limited quantities, and ^212^Pb production requires robust separation methods [133]. However, several companies are working towards scaling up the supply of various α-emitters. For example, one research group recently described a novel method for producing high yields of ^212^Pb using a single-chamber generator based on decaying ^224^Ra or ^228^Th [134]. In addition, this generator is compact and user-friendly, making it a key candidate for use at a nuclear medicine facility [134]. With respect to ^225^Ac, future production methods are underway for improving the yield, and in fact, large-scale production of ^225^Ac through cyclotron proton irradiation of ^226^Ra has shown promise [135]. Meanwhile, ^227^Th has been commercially available for many years and it can be produced in virtually unlimited amounts with current technology, making it a viable radionuclide for several forms of TαT [133]. Therefore, although the production of some α-emitting radionuclides remains limited, there is strong evidence supporting the suggestion that capacity will increase as clinical results promoting the benefits of TαT continue to grow and technology advances.

## 10. Future Perspectives

Nb-based TαT is an emerging therapeutic and diagnostic field with great potential. Although TαT has primarily been investigated for cancer therapy, a future application of this therapy could be explored for other diseases, including, but not limited to, infections, autoimmune disorders, some neurological conditions, vascular diseases, or even pain management. For example, in the case of Alzheimer’s or Parkinson’s disease, an accumulation of toxic proteins or misfolded aggregates is often involved in disease pathogenesis. Targeting these pathological protein aggregates with radiolabeled Nbs could offer a novel therapeutic approach. Another promising direction of Nb-based TαT is the development of multivalent Nbs. Currently, most Nb-based TαT approaches involve monovalent antibodies that target a single antigen. Multivalent nanobodies could increase treatment specificity and potency, especially for tumors with heterogeneous or multiple target antigens. In fact, bispecific bivalent Nbs have already shown promise in Nb therapy, demonstrating the potential to radiolabel such Nbs for targeted delivery of radionuclides [136]. Although the current review has highlighted a few studies where Nbs were used in combination with other treatment modalities, combining Nb-based TαT with other treatment regimens, such as chemotherapy, immunotherapy, or radiotherapy, should continue to be explored. By synergistically targeting multiple pathways or utilizing complementary mechanisms of action, combination therapies could significantly enhance treatment efficacy and overcome resistance.

## 11. Conclusions

In conclusion, naturally occurring Nbs, or sdAbs, are favorable and versatile tools for a variety of biomedical applications. With Nbs’ small size and strong antigen-binding abilities, they are able to access and bind to antigen cavities on tumor cells that are otherwise inaccessible to full-length mAbs. Moreover, the improved stability, low ‘off-target’ effects, and decreased immunogenicity allow for better clinical applications, with reduced risks to patients. Importantly, the conjugation of Nbs targeting specific cancer markers to radionuclides has proven successful for both diagnostic and therapeutic purposes, specifically TRNT. The use of α-emitters in TRNT has provided further advantages, with the high LET of α particles and shorter half-lives, to better match those of Nbs. As a result, and evidenced by many pre-clinical studies, the current review highlights that Nb-based TαT holds great potential for future cancer therapeutic applications.

## Figures and Tables

**Figure 1 cancers-15-03493-f001:**
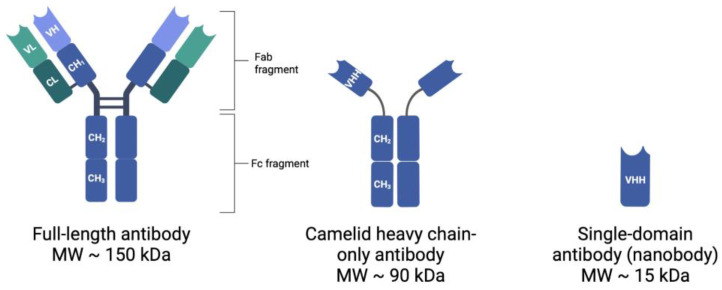
Comparison of full-length antibody and nanobody structures.

**Figure 2 cancers-15-03493-f002:**
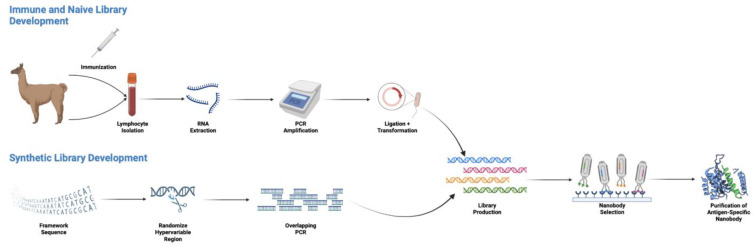
Production of antigen-specific figure single-domain antibodies (sdAbs), or nanobodies.

**Table 1 cancers-15-03493-t001:** Summary of the advantages of nanobodies.

Advantage	Molecular Reason
High stability	Ability to refold after denaturation; resistance to acidic and alkaline conditions
Improved antigen access	Small size and presence of large exposed loop to penetrate antigen cleft
Specific antigen binding	Greater structural variability
Low off-target toxicity	Small size and quick blood clearance
Rapid tumor penetration	Small size
Low immunogenicity	High degree of homology to human VH
Facile production	Only one mature domain required

**Table 2 cancers-15-03493-t002:** Actively recruiting clinical trials involving radiolabeled Nbs for diagnostic and treatment purposes. Data obtained from the National Institutes of Health clinical trials database, ClinicalTrials.gov [94].

Target	Nanobody	Disease	Primary Purpose	Clinical Trial	Phase
HER2	^68^GaNOTA-Anti-HER2 VHH1	Breast Neoplasm	Diagnostic	NCT03331601	2
		Breast Carcinoma			
MMR	^68^GaNOTA-Anti-MMR-VHH2	Malignant Solid Tumor	Diagnostic	NCT04168528	1/2
		Breast Cancer			
		Head and Neck Cancer			
		Melanoma (Skin)			
MMR	^68^GaNOTA-Anti-MMR-VHH2	Squamous Cell Carcinoma of Head and Neck	Diagnostic	NCT04758650	2
		Cancer			
		Carotid Stenosis			
		Atherosclerosis of Artery			
		Hodgkin Lymphoma			
		Non-Hodgkin Lymphoma			
		Hemophagocytic Lymphoistiocytosis (HLH)			
		Cardiac Sarcaoidosis			
HER2	^99^mTc-MIRC208	Cancer	Diagnostic	NCT04591652	N/A
HER2	^99^mTc-NM-02	Breast Cancer	Treatment	NCT04674722	Early Phase I
	^188^Re-NM-02				
PD-L1	^99^mTc-NM-01	Non-Small Cell Lung Cancer	Diagnostic	NCT04992715	2

**Table 3 cancers-15-03493-t003:** Properties of α-emitting radionuclides [100,101].

Parent	α-Emitting Daughters	T_1/2_	Energy of Emitted Particle (MeV)
^211^At		7.2 h	6
	^211^Po	516 ms	7.5
^225^Ac		9.9 d	6
	^211^Fr	4.9 min	6
	^217^At	32.3 ms	7
	^213^Bi	45.6 min	6
	^213^Po	3.7 µs	8
^227^Th		18.7 d	6
	^223^Ra	11.4 d	6
	^219^Rn	4 s	7
	^215^Po	1.8 ms	7.5
	^211^Bi	2.2 min	7

**Table 4 cancers-15-03493-t004:** Summary of preclinical studies investigating targeted radionuclide therapy using nanobody delivery vehicles.

Target	Radionuclide	Labeling Strategy	Model Used	Main Findings	Reference
HER2	^131^I (β- and γ-emitter)	Prosthetic group SGMIB	In vitro: HER2^+^ cell lines: BT474/M1, JIMT-1, SKOV-3, and SKOV-3.IP1	High tumor uptake in both mouse models, and low normal tissue uptake	[76]
			In vivo: HER2^+^ tumor xenograft mouse models: (1) BT474/M1 and (2) SKOV-3.IP1	[^131^I]SGMIB-2Rs15d alone, or in combination with trastuzumab, significantly extended tumor survival	
CD20	^177^Lu (β-emitter)	DTPA conjugation	In vitro: Daudi (hCD20^pos^), Reh (hCD20^neg^), and murine B16-F10 cell lines; hCD20^pos^ B16 cell line generated for study	^177^Lu-DTPA-sdAb 9079 showed much lower absorbed doses in non-target organs compared to ^177^Lu-DTPA-rituximab	[106]
			In vivo: C57BL6 and CB17 SCID mice bearing hCD20^+^ B16 tumors	^177^Lu-DTPA single-domain antibody (sdAb) 9079 resulted in significantly higher survival rates compared to control	
HER2	^177^Lu (β-emitter)	DTPA conjugation	In vitro: SKOV3 and SKOV3-LUC (in-house HER2^pos^/Luciferase^pos^)	Unwanted kidney retention of radiolabeled nanobodies was reduced when using untagged nanobodies and co-infusion with Gelofusin	[107]
			In vivo: Female athymic mice bearing HER2^+^ (SKOV3) tumors	^177^Lu-DTPA-2Rs15d efficiently inhibited tumor growth	
PSMA	^213^Bi (α-emitter)	DOTA conjugation	In vitro: PSMA-expressing LNCaP cells	^213^Bi-labeled nanobodies induced DNA double-strand breaks in both in vitro and in vivo models	[108]
			In vivo: LNCaP xenograft BALB/C mice		
HER2	^211^At (α-emitter)	Prosthetic group SAGMB	In vitro: HER2^+^ BT474M1 breast carcinoma cells	Anti-HER2 sdAb 5F7 can be efficiently labeled with ^211^At with excellent affinity and immunoreactivity	[75]
			In vivo: SCID mice with subcutaneous BT474M1 xenografts	[^211^At]SAGMB-5F7 had high and prolonged tumor targeting and rapid normal tissue clearance, with iso-[^211^At]SAGMB-5F7 demonstrating even more favorable results	
HER2	^225^Ac (α-emitter)	DOTA conjugation	In vitro: SKOV-3 and MDA-MB-231 (low HER2-expressing) cells	HER2 nanobody 2Rs15d can be effectively labeled with ^225^Ac with preserved affinity and immunoreactivity	[115]
			In vivo: SKOV3 tumor-xenografted mice	^225^Ac-DOTA-Nb was cytotoxic in vitro in a HER2-dependent manner and quickly accumulated in HER2^+^ tumors in vivo	
				Renal accumulation of ^225^Ac-DOTA-Nb was effectively reduced with co-infusion of Gelofusin	
HER2	^211^At (α-emitter)	Conjugation with three different coupling reagents: SAGMB, SAB, MSB)	In vitro: SKOV3 cells	Nanobody labeled via SAGMB ([^211^At]SAGMB-2Rs15d) was deemed the preferred agent as the biological properties best matched the physical characteristics of ^211^At	[112]
			In vivo: SKOV3 tumor-xenografted female nude BALB/C mice	[^211^At]SAGMB-2Rs15d showed fast and high accumulation in a HER2^+^ tumor mouse model together with a low non-target organ uptake	
HER2	^225^Ac (α-emitter) and ^131^I (β-emitter)	^225^Ac: DOTA-based conjugation	In vitro: HER2^+^ cell lines SKOV3.IP1 and MDA-MB-231Br	[^131^I]-2Rs15d and [^225^Ac]-2Rs15d both showed high and specific tumor uptake in HER2^+^ brain lesions	[113]
		^131^I: prosthetic group SGMIB	In vivo: female athymic nude mice (Crl:NU(NCr)-Foxn1^nu^) with SVOV3.IP1or MDA-MB-231Br tumor xenografts	Administration of radiolabeled nanobodies alone and in combination with trastuzumab significantly increased median survival in tumor models (that were unresponsive to trastuzumab alone)	
HER2	^213^Bi (α-emitter)	DTPA conjugation	In vitro: SKOV-3 (HER2^+^) and CHO (HER2^−^) cell lines	[^213^Bi]-DTPA-2Rs15d demonstrated a high tumor uptake, but low uptake in normal tissue (co-infusion of gelofusine also led to 2-fold reduction in kidney uptake)	[114]
			In vivo: athymic nude mice (Crl/NU(NCr)-Foxn1^nu^) with SKOV3 tumor xenografts	[^213^Bi]-DTPA-2Rs15d alone and in combination with trastuzumab significantly increased median survival in in vivo model	
CS1	^225^Ac (α-emitter)	DOTA conjugation	In vitro: 5T3MMvt and 5TGM1 GFP^+^ cells	Administration of anti-CS1 sdAbs radiolabeled with ^225^Ac resulted in significantly increased survival of mice, an increase in CD8+ T-cells, and more PD-L1 expression on immune and non-immune cells	[116]
			In vivo: C57BL6 mice injected with 5T3MM or 5TGM1 cells		
5T33 idiotype	^177^Lu (β-emitter) and ^225^Ac (α emitter)	DTPA conjugation for ^177^Lu, DOTA conjugation with ^225^Ac	In vitro: 5T3MM cells	Radiolabeled anti-idiotype sdAbs significantly delayed tumor progression in mice with low 5T33 myeloma lesion load	[117]
			In vivo: C57BL/KalwRij mice intravenously injected with 5T3MM cells and C57BL/6 mice	Membrane expression of paraprotein was confirmed in five out of seven patients with newly diagnosed myeloma, and two anti-idiotype sdAbs were successfully generated from serum-isolated paraprotein	
HER2	^211^At (α-emitter)	Prosthetic group SAGMB	In vitro: BT474 cells	Clonogenic survival of BT474 cells exposed to *iso*-^211^At-SAGMB-5F7 was reduced	[118]
			In vivo: NOD-scid-IL2Rgamma^null^ and athymic mice with subcutaneous BT474 xenografts	Dose-dependent tumor growth inhibition was observed with ^211^At-labeled anti-HER2-specific nanobodies 5F7 and VHH_1028; prolongation in median survival was over 400% for both nanobodies.	
CD20	^225^Ac (α-emitter)	DOTA conjugation	In vivo: C57BL/6 mice subcutaneously transplanted with B16 melanoma cells expressing human CD20	^225^Ac-DOTA-9079 (nanobody targeting huCD20) resulted in delayed tumor growth and increased blood levels of various cytokines	[119]
				^225^Ac-DOTA-9079 also promoted an environment for antitumoral immune cells and increased the percentage of programmed death-ligand 1 (PD-L1)-positive immune cells in the tumor microenvironment	
HER2	^131^I (β- and γ-emitter)	Iodogen method	In vivo: BALB/c mice subcutaneously injected with MDA-MB-231 (HER2^−^) or SKBR3 cells (HER2^+^)	^131^I-NM-02 was effectively taken up by HER2^+^ tumors with rapid blood clearance and favourable biodistribution	[120]
				^131^I-NM-02 significantly inhibited tumor growth and extended survival	
PD-L1 and cytotoxic T-lymphocyte-associated protein 4 (CTLA-4)	^131^I (β- and γ-emitter)	Iodogen method	In vitro: B16F10 and MCF-7 cells	^131^I-KN046 demonstrated high affinity and specificity for PD-L1/CTLA-4 immune targets and strong intratumoral retention capability	[121]
			In vivo: female BALB/c mice injected with B16F10 or MCF-7 cells	^131^I-KN046 enhanced the immune response, leading to upregulated expression of MHC-1 and Fas surface molecules, increases in T-cell activation, and a greater number of tumor-infiltrating immunocytes	
PD-L1	^131^I (β- and γ-emitter)	Chloramine-T method	In vitro: H460 (PD-L1^+^) and A549 (PD-L1^−^) cell lines	H460 cells demonstrated high ^131^I-Nb109 uptake	[122]
			In vivo: female BALB/c nude mice inoculated with H460 cells	^131^I-Nb109 showed accumulation in H460 tumors, successfully inhibited tumor growth without toxic side effects, and induced H460 cells to release DAMPs	

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
