# Peer review of "Targeted Alpha Therapy (TAT) with Single-Domain Antibodies (Nanobodies)"

_cancers, 2023, doi:10.3390/cancers15133493_

Round 1

Reviewer 1 Report

The topic of this review is not new. Besides, there are very recent published reviews on this topic, with better figures and explanations of various aspects regarding Nanobiotics (e.g., https://doi.org/10.3390/ijms24065994, https://doi.org/10.3389/fimmu.2023.1012841). However, the two recent included studies (2023) added value to this manuscript. Table 2 is of high quality. I have listed some suggestions for consideration below:

1. There are many redundant data in the text. I suggest the authors to carefully read their paper and remove them. Just one example:

- Lines 97-99: “Although TαTs using mAb vehicles have shown some success in clinical trials, they are limited by complications such as myelosuppression and abnormal liver function [20,21].”

- Lines 177-179: “mAbs” …”can result in off-target toxicity, especially in the liver”.

2. Also, there is a lot of data mentioned both in the text and Table 2. Please keep data in the table and shorten it considerably from the text.

3. Some info in the text could be shortened, as it is well known (e.g., the paragraph “Full-length Antibodies” etc).

4. Please divide “Future perspectives” and “Conclusion” in two separate paragraphs. In “Future perspectives”, please write real ones, not what was already reported and published (references 123-131).

5. Please insert ongoing studies with Nanobodies, from clinicaltrials.gov, in order to bring more novelties.

6. References: please double check them, as some of them are similar (e.g., references 20 and 104). More data from 2023 could be added.

The quality of the English language is generally good. Only some minor changes are required, including shortening the long sentences.

Reviewer 2 Report

Targeted radionuclide therapy is a promising tumor treatment method. This article gives a comprehensive description of the targeted alpha therapy (TαT) based on nanobodies. There are some suggestions, listed as follows:

1.     As a review of TαT based on nanobodies. I think you need a table to summarize the advantages of nanobodies compared with traditional full-length antibodies, but it is not necessary to spend a lot of space to introduce the basic properties and screening process of antibodies, and some properties and advantages of nanobodies have been repeatedly mentioned in many places in the article.

2.     Some data needs to be updated

L37: “Cancer is one of the deadliest diseases in the world, accounting for nearly 10 million deaths worldwide in 2020 alone”

L45: “Since then, 41 monoclonal antibodies (mAbs) have been approved by the FDA for clinical use against cancer, as of 2021”

3.     Whether there are clinical cases of TαT based on nanobodies, if so, it should be summarized in the table, and explain the clinical stage and efficacy.

4.     Whether the production and preparation of α-Emitting Radionuclides will become the limiting factor of its industrialization?

5.     There is no introduction about the linker for radiolabeling Nbs with α emitters in this article.

Round 2

Reviewer 1 Report

I am very pleased after reading the revised version of the manuscript. Now, it looks professional, nicely structured and developed, while it also brings many recent and accurate data. I support its publication in the present form.

Thank you

Reviewer 2 Report

Thank you for submitting an updated version of the manuscript, the manuscript has been improved significantly. I recommend this review for publication.